# DMA Study of the Molecular Structure of Porcine Fat in-Water Emulsions Stabilised by Potato Starch

**DOI:** 10.3390/ijms22147276

**Published:** 2021-07-06

**Authors:** Ryszard Rezler

**Affiliations:** Department of Physics and Biophysics, Faculty of Food Science and Nutrition, Poznań University of Life Sciences, Wojska Polskiego 38/42, 60-637 Poznań, Poland; ryszard.rezler@up.poznan.pl

**Keywords:** complex index, emulsion, porcine fat, potato starch, rheology

## Abstract

The aim of the study was to determine how the molecular structure of porcine fat-in-water type emulsions stabilised with potato starch affected their rheomechanical properties. Dynamic mechanical analysis (DMA) and instrumental analysis of the texture were the method used in experiments. Starch gels with concentrations corresponding to the water starch concentration of the examined emulsions were used as control systems. The analysis of the starch and starch–fat systems showed that the values characterising their rheomechanical and textural properties reflected the spatial reaction of the amylose matrix to dynamic mechanical interactions. Changes in their values resulted from conformational changes in the structure of segments and nodes of the lattice, conditioned by the concentration of starch and the presence of fat. As a result of these changes, starch–fat emulsions are distinguished by greater densities of network segments and nearly two times greater functionalities of nodes than starch gels. The instrumental analysis of the texture showed that the values of the texture parameters in the starch gels were greater than in the starch–fat emulsions. The high values of the correlation coefficients (R~0.9) between the texture determinants and the rheological parameters proved that there was a strong correlation between the textural properties of the tested systems and their rheomechanical properties.

## 1. Introduction

The fat phase constitutes an important component of restructured meat products designed following traditional recipes [1]. An extraordinary functional importance of fat results, among others, from its effect upon the product texture, juiciness and palatability [2,3]. Fat in food products reacts with aromatic substances resulting in their sensory balancing and, therefore, it provides the desirable texture, quality parameters, and shelf-life. The increasing demand for low-fat products observed in recent years forces manufactures to modify product formulations, among others, by reducing fat content and replacing it, for example, by water. When the extent of fat replacement exceeds 70%, which occurs in the case of typically low-fat products, the continuous phase of the product is transformed into oil-in-water type of emulsion losing simultaneously rheological features characteristic for condensed emulsions [4]. This leads to considerable changes in the product quality parameters, in particular, texture and water binding capability [5,6]. That is why fat replacement is achieved by the introduction into the product of water in combination with other additives which do not only bind water but which are also characterised by structure-forming properties. Such characteristics are attributed, in particular, to hydrocolloid preparations obtained on the basis of proteins, starch as well as other polysaccharides [7,8]. From among all polysaccharides, starch—due to its physico-chemical properties—is one of the most important structure-forming food constituents.

This refers both to potato starch as well as to starch obtained from cereals. Starch in the presence of water, and at elevated temperature, forms permanent spatial structures. For this sake, if used as an additive, it can modify the structure and texture of food products [9,10]. Starch gelling properties or of its isolated constituents are utilised not only in production of various gelatine desserts, fruit mousses, sauces, and bread baking but also as a fat replacement in the manufacture of cured meat products.

Starch—in particular, its amylose fraction—is capable of forming complexes with various organic and inorganic compounds (among others, with iodine, alcohols, free fatty acids, emulsifiers, monoacylglycerols as well as other surface-active substances) in whose presence amylose assumes the form of a spiral and then it forms a crystalline structure [11]. Inclusion complexes can modify starch properties and functionality; for example, they can restrict its solubility in water inhibiting processes of retrogradation and enzymatic hydrolysis [12]. That is why so many experiments are focused on issues associated with gelation and starch retrogradation in the presence of fats [13,14]. The majority of them concern monoglycerides or single fatty acids.

Among articles connected with the impact of fats on starch system properties, the majority concern fats of plant origin. Few researchers investigated interrelationships between starch systems and animal-derived fats, especially as the structure of fatty acids derived from animals is very complex in comparison with plant fatty acids. This refers not only to the content of saturated fatty acids, which clearly prevail in animal fats, but also to their molecular structure. If such investigations are undertaken, then the majority of them are focused on issues associated with structure stabilisation of the continuous phase in low-fat food articles confining themselves to investigations of final product properties.

The rheological properties of materials, such as elasticity, strength, and plasticity, are manifested under short-term mechanical stresses [15,16]. They refer to the relationship between stress and deformation and their changes over time. The values of rheomechanical determinants found in measurements are generally very significantly and specifically related to the texture of food products. Texture is basically defined as a set of organoleptic characteristics related to food products [17,18]. It includes a set of physical characteristics resulting from the structure of these products, belonging to their mechanical and rheological properties and being a complex quantity. As results from this definition, rheomechanical properties are an integral part of the concept of texture and its significant determinants.

Therefore, parallel analysis of rheological and textural properties has been the subject of many studies, including those investigating how the substitution of starch for fat affects the rheological and textural properties of meat products. Barbut [19] identified the influence of regular and modified potato and maize starches on frankfurter type products. Campo and Tovar [20] conducted a study to investigate how the content of starch improved the viscoelastic properties of meat gels. Shon and Chin [21] investigated the effect of myofibrillar protein mixed with tapioca starch on model pork sausages.

So far there have been few articles on the determinants of molecular mechanisms responsible for the rheomechanical and thus the textural properties of emulsified low-fat meat products. There have been even fewer studies on the interrelations between changes in the molecular structure of starch-, fat-, and protein-containing systems and their rheomechanical properties. Available studies only show that the values characterising the rheomechanical and textural properties of emulsified sausages reflect the reaction of the spatial protein matrix to the mechanical interaction and the physical state of the meat emulsion being the continuous phase of the system [22,23,24]. The rheomechanical properties in modified sausages are determined not only by the sol-gel transformations of proteins and the gelation of starch being a fat substitute, but also by the interaction between proteins, starch, and fat [25,26].

In view of the circumstances discussed above, extensive research was undertaken to investigate the structure-forming and water-binding effects resulting from the introduction of native as well as physically and chemically modified potato starch into the animal (porcine) fat emulsion system in water, modelling the continuous phase in low-fat comminuted meat products. The results presented in this study are part of this research. The aim of this study was to determine the influence of the molecular structure of starch–water–animal (porcine) fat systems on their rheomechanical and textural properties. The model of the continuous phase used in the low-fat comminuted meat products was not complete because it did not contain proteins. The results concerning this model will be the subject of another publication.

## 2. Results and Discussion

### 2.1. Rheomechanical Properties

In the case of poorly cross-linked starch and starch–fat systems, the delicate network of intermolecular links guarantees the shape stability of gels in conditions of weak non-destructive mechanical interaction. Oscillation rheometry expanded to the technique of dynamic-mechanical analysis (DMA) is one of the most universal and objective methods which can be applied in investigations of these types of systems, since it affords possibilities of determining values of elasticity coefficients, coefficients of internal friction (viscosity) as well as other rheomechanical parameters at any frequency of non-destructive interactions of the developed system structure. The storage modulus (*G′*), loss modulus (*G′′*), and loss tangent (*tgδ*) are particularly important rheomechanical parameters. The storage modulus is related with the part of the potential deformation energy which is retained during periodic deformations. The loss modulus is related with the part of energy dissipated as heat. The loss tangent is a measure of internal friction; it determines the relative amount of energy dissipated in the material during one deformation cycle. These parameters are often used to correlate the rheomechanical and textural sensory properties of various products to which starch is added to improve their textural properties.

This particularly applies to various types of emulsified low-fat meat products in which starch is a fat substitute, especially cold, processed types of meat emulsions, such as various types of sausages or pâtés [27,28,29]. In view of this, to determine the influence of the molecular structure of starch–water–animal fat systems on their rheomechanical properties, changes in the kinetics of the basic parameters determining these properties were analysed.

With the passage of time counted from the moment of attaining the temperature of surroundings at different starch concentrations, the research results obtained with the assistance of the DMA technique of the experimental starch gel (starch concentrations: 0.25; 0.21; 0.17; 0.14; 0.12 and 0.06 g/g) and starch–fat emulsion (starch concentrations: 0.2; 0.17; 0.13; 0.11; 0.09; 0.05 g/g) systems are illustrated by model runs of dependences of the storage modulus *G*′(*t*) presented in Figure 1.

The initial values different from zero of the dynamic rigidity modulus *G_o1_* indicate that already during the cooling process a spatial network developed in the examined systems with the concentration of *n_os_(c_s_)* segments dependent on the concentration of starch (*c_s_*). The approximate correlation between the storage modulus of highly elastic polymer networks and the concentration of segments in these networks is determined by equation in the following [30] (Equation (1)):(1)$$n_{s}\;\frac{G^{\prime }}{RT}$$

The increase of hydrogel stiffness moduli and of the examined emulsions observed with the passage of time accompanied by different starch concentrations provides evidence about the fact that the concentration of effective segments in this network increased. It is also worth noting that the kinetics of the course of the crosslinking processes conforms to the description of the Avrami type equations [31] (Equation (2)):(2)$$n_{s}\left(t\right)=n_{os}+\left[n_{ws}-n_{os}\right]\left\{1-\exp\left[-{\left(kt\right)}^{m}\right]\right\}$$
where, *n_ws_* and *n_os_* designate, respectively, initial and final concentrations of network segments; *k* is the kinetics constant and m is the power exponent generally related to crystallite morphology [32]. Following fitting of Equation (2) with the data obtained with the assistance of the DMA method, the power exponent of time *m* assumes values dependent on the starch concentrations (*c_s_*) in the examined systems. In emulsions: *m* ≅ 1 for 0.17 g/g < *c_s_* < 0.2 g/g and *m* ≅ 2 for *c_s_* < 0.17 g/g, whereas in starch gels of starch concentrations corresponding to the concentration of the starch in water of the examined emulsions: *m* ≅ 1 at the interval 0.14 g/g > *c_s_* < 0.25 g/g and m ≅ 3 for *c_s_* < 0.14 g/g. The nature of changes of the *k* constant (Figure 2) indicates diversification of molecular processes determining the extent and formation velocity of the spatial network nodes in gel and emulsion systems.

The diagrams (Figure 3) indicate that the equilibrium *n_ws_* and initial *n_os_* runs of network segment concentrations undergo linearization in the *c_s_*^2^ function and assume a slope depending on the condition of the system and concentration interval.

The values are expressed as means ± SD (*n* = 3). This indicates diversification of the dispersive structure of starch gels and starch–fat emulsion developing as a result of starch retrogradation.

Bearing in mind the fact that segment concentration in the gel network is determined by the product of the node concentration (*w*) and the functionality of these nodes (*f*):(3)$$n_{s}=0.5fw$$

Equation (3) can be presented in the following form:(4)$$n_{s}\left(t\right)=0.5w\left\{f_{o}+\left(f_{k}-f_{o}\right)\left[1-\exp\left[-{\left(kt\right)}^{m}\right]\right]\right\}$$

It is evident from this dependence that, at a given starch concentration in the system and at constant temperature, the increase of gel stiffness in the time function is caused by the buildup of nodes which consists in the binding of new segments of macromolecules leading to the enhancement of average functionality of the set of nodules stabilizing the spatial network. It is apparent from the obtained node functionality values of equilibrium network as well as from the ratio of directional coefficients of anamorphoses (Figure 3) that the initial functionality means of network nodes for the two examined systems assume comparable values in the entire range of starch concentration *c_s_*: for starch gels and emulsions *f_o_* ≅ 7.9 and 8, respectively. This means that the initial spatial structure of the two examined systems is determined by nodes formed by at least two tetrafunctional structures which are probably bispiral associations of chains. This provides evidence that already during the cooling process of the examined systems a spatial network is formed of structure parameters dependent on the polymer concentration. Together with aging of the gels and emulsions, this network undergoes gradual reconstruction involving, mainly, the increase of the functionality of initiating nodes in the result of adsorption of successive segments of macromolecules. This is confirmed by high values of mean final functionalities *f_k_* ≅ 120 in starch gels and *f_k_* ≅ 220 in emulsion systems. These high values of mean final functionalities *f_k_* make it possible to assume that the structure of spatial network of the examined systems is determined by nodes which are hexagonal associations of bispiral structures. Earlier, on the basis of measurements of dynamic storage modulus *G’* (Figure 1), it was found that during the cooling process, a spatial network with different segment concentration was formed in the examined gels and emulsions.

The spatial network of such systems is the result of connections between different fragments of macromolecules neighbouring each other. These connections are formed in starch by being spiralised (mainly amylose) and merged into bihelical associates. This form is characteristic of native and retrograded starch.

On the other hand, during gelatination and, later on, gelation of starch in the presence of fat, additionally with part of amylose, fatty acids develop complexes in the form of single helixes [33,34]. The second constituent of starch—amylopectin—is only slightly involved in the development of complexes because its side chains are not long enough to form complexes with fatty acids. Another significant obstacle preventing the formation of complexes is the spatial structure of amylopectin (densely branched side chains) [35]. Table 1 presents *CI* (complex index) on the basis of the reduction in the iodine binding capacity of amylose [36]. The *CI* value increases together with the decrease of starch content in systems (water and fat content on all systems is the same) (Table 1).

Porcine fat is mostly composed of long-chain fatty acids (C16, C17 and C18). It is clear from chromatographic investigations (Table 2) that saturated fatty acids (C16, C17 and C18) as well as unsaturated fatty acids (C16:1, C18:1 and C18:2) are prevailing constituents making up approximately 42.2% and 51.3%, respectively, of all fatty acids. Therefore, it can be assumed that the rheomechanical properties of the examined emulsions are affected, first and foremost, by these three dominating fatty acids (Table 2).

Generally speaking, the *CI* value was affected by the structure of fatty acids. On the one hand, the fatty acids containing a long carbon chain formed complexes with amylose less effectively because of their amphiphilic nature, dispersion states, and steric hindrances [36]. On the other hand, the *CI* values also tended to increase as the number of double bonds in the fatty acid increased [36,37].

Bearing in mind the complex structure of the applied fat, it is not possible to determine unequivocally which of the fatty acids exerted the strongest impact on the process of formation of amylose–fat complexes. The presence of starch–fat complexes in the range of high starch concentrations (up to *c_s_* = 0.17 g/g) did not result in significant differences in the retrogradation process of the two examined systems (emulsions and gels). This was confirmed by the time power exponent *m* which was earlier determined from the fitting of Equation (2) for the data obtained with the assistance of the DMA method. The value of this exponent (*m* ≅ 1) indicates that the re-crystallisation of the examined systems occurred at the initial nucleation phase of crystalline form nuclei.

The ultimate spatial structure of starch gels is the result of expansion of the already existing network segments and the development of new segments formed following the entanglement of individual starch chains within the amorphous fraction and the inclusion into crosslinking of short amylopectic branches.

On the other hand, in the case of starch–fat emulsions, the additional increase in the degree of crosslinking is the outcome of development of bonds between amylose–fat complexes. This was reflected in greater values of both equilibrium concentrations of network segments (Figure 3) as well as mean terminal functionalities *f_k_* (in emulsions: ≅220, in starch: ≅120).

At high starch content, polymer–polymer interactions dominated in the examined systems with polymer–fat ones observed much less frequently. This explains the determined similarity in the spatial structure of starch emulsions and gels. It was only when the starch content *c_s_* < 0.17 g/g, that the interactions polymer–fat began to predominate and the molecular structure of starch–fat complexes was found to determine, in a significant manner, the spatial structure of starch gels in emulsions.

In recent years, investigations on the molecular structure of starch–fat complexes have been carried out quite intensively by many researchers [38,39]. The results of these studies suggest that the complexes were organised in lamellae, packed in microscopic spheroids, and clustered in aggregates. Moreover, the complexes of saturated fatty acids within the lamellae formed more ordered structures than the complexes of unsaturated acids [40,41]. Uncomplexed fatty acid ligands were distributed in the lamellae formed by the starch–fat complexes [33]. In emulsion systems, within the whole range of the starch content below *c_s_* < 0.17 g/g, the exponent of time was *m* = 2. This suggests that the starch chain associates are mostly well-ordered. They are partially organised in lamellar structures. This organisation system may be largely forced by the molecular structure of starch–fat complexes and by uncomplexed ligands of fatty acids [42,43].

Such starch aggregate conformation in emulsions is additionally favoured by the fact that porcine fat crystallizes below starch gelation temperature. Hence, its solidification takes place within the developing gel network which consolidates its structure even more and confines further starch crystallisation.

The time power exponent, in starch gels for starch concentration of *c_s_* < 0.14 g/g, exhibits the value of *m* ≅ 3 indicating that three-dimensional crystalline structures were developing in them. Starch aggregates assume conformations of a statistical bundle. Due to steric reasons, such conformation of starch associates was impossible at higher starch contents in gels. On the other hand, in starch–fat emulsions, the existing lamellar structure and crosslink bonds between amylose–fat complexes associated with it, made the helix-bundle transfer impossible.

Starch–water–fat systems form dispersions. Amylose granules dissolved in water constitute the solid phase, whereas swelled granule residues together with fat form the dispersed phase. Rheological properties of the emulsion depend on many factors, among others, on the size of the fraction in continuous and dispersed phases as well as the interactions between them. Mechanical properties of individual phases determine values of rheological parameters of the examined systems [36]. Spatial systems in starch–fat emulsions in comparison with starch gels distinguish themselves by a greater density of network segments and almost twofold greater node functionality. However, from the point of view of morphology, the node structure in both systems is similar; they are made up of complex aggregates of starch chains. Despite this, segments of the spatial network in emulsions distinguish themselves by a lower binding energy in comparison with starch gels. The outcome of this is variation in stiffness and dissipativeness of their equilibrium spatial systems. This finds its expression both in higher values of storage moduli *G’* (Figure 1) as well as in the ability for the emulsion energy dissipation *tgδ* (Figure 4) in comparison with starch gels. As a result of this, the response to dynamic mechanical actions of starch–fat emulsions is characteristic for bodies characterized by visco-plastic properties, in contrast to starch gels which behave like visco-elastic solid bodies.

As mentioned above, the effect of polymer–fat starts to prevail in the starch–fat emulsion’s settings with the following starch content *c_s_* < 0.17 g/g. Molecular structure of complexed fatty acids and part of non-complexed fatty acids significantly determine their rheomechanical properties. The depletion of the starch content in the emulsion’s setting contributes to the weakening of their spatial systems. As a result, fat is separated from a gel matrix. The processes of kinetics fat storage modulus, which are presented in Figure 1, reveal relaxation processes, which probably relate to its crystallization. The influence of fat crystallization on rheomechanical properties of starch–fat emulsions is particularly visible in the settings with the lowest starch content (*_Cs_* = 0.05 g/g). As a result, the setting’s ability to energy dissipation rises (Figure 4). With slightly higher starch content, the effect of fat crystallization was masked by starch gel matrix’s elastic response.

### 2.2. Texture

Table 3 shows the results of the analysis of the texture of starch gels and starch–fat gels with pork fat, conducted after one day and seven days of storage at room temperature. The native starch gels were characterised by higher values of the texture parameters than the starch–fat gels. The only exception was the samples with the lowest concentration of starch.

It is likely that the fat additive reduced the values of the surface destructive force, hardness, consistency, cohesiveness, and viscosity coefficient. The storage time was another factor that significantly influenced the texture. The values of the mechanical parameters of the samples analysed on the first day of storage were lower than those of the samples stored for seven days at room temperature.

One of the main problems in the analysis of the results of texture measurements is the lack of rational nomenclature describing these parameters. This is because texture parameters are complex quantities reflecting various physical phenomena occurring during the measurement. They describe changes in the elastic and surface properties as well as the internal structure of the systems under analysis [44]. The analysis of the starch and starch–fat systems showed that the values characterising their mechanical and rheological properties reflected the spatial reaction of the amylose matrix to dynamic mechanical interactions. These values changed as a result of conformational changes in the structure of segments and nodes of the spatial network, conditioned by the concentration of starch. Therefore, it is possible to assume that their rheomechanical and textural properties are interrelated.

As was mentioned above, one of the main problems in texture analysis is the lack of definitions of the tested parameters clearly related to the rheomechanical quantities. So far none of the attempts has provided satisfactory results [45,46,47]. However, it is possible to check the strength of the relation between selected rheomechanical properties and textural parameters. In our study linear (Pearson) correlation was used to check the relation between the hardness and the storage modulus *G*’ and between the viscosity coefficient and the loss modulus *G*′′ (*G*′′= *tg**δ x G*′) after 24 h and after 7 days of storage (Table 3).

The high values of the correlation coefficient (respectively *R* ≥ 0.89 and *R* ≥ 0.91) indicated a strong correlation between the textural and rheomechanical properties of the tested systems. The results of the analysis of the starch and starch–fat systems let us assume that the main factors affecting the texture and rheomechanical properties of starch gels are: the density of the continuous phase spatial lattice formed by amylose diffusing from starch granules during pasting, swollen, pasted granules consisting of amylopectin as well as their interactions [48]. These factors depend on the starch concentration.

Apart from the aforementioned factors referring to the starch gels, the texture of the starch–fat gels was mainly affected by the interaction of fatty acids with amylose in the form of amylose–fatty complexes and by non-complexed fat, which was inside or outside the spatial lattice, depending on the starch concentration. Like the starch gels, the values of the textural parameters of the starch–fat gels varied depending on the starch concentration.

The hardness and viscosity coefficient were deliberately chosen for analysis because these two texture parameters best described the elasticity and viscosity of the tested systems. The measurement of hardness consisted in forcing the indenter into the tested material beyond the elasticity limit until permanent deformation was observed. Therefore, hardness can be defined as a measure of material’s resistance to permanent deformation. The value of this parameter is determined by the internal structure of the material and it is most closely related to its elastic properties [49]. Hardness is directly related to the surface destructive force, which determines the yield point.

The viscosity coefficient is not the equivalent of dynamic viscosity, but is related to it. Consistency and cohesiveness are related to it. This quantity was correlated with the loss modulus *G*′′, which is often referred to as the dynamic viscosity modulus. It indicates the energy of the sample converted to heat and is an indicator of its viscous or liquid behaviour [30].

The increase in the values of the texture parameters after seven days of storage may have been caused by changes resulting from the retrogradation of amylose and partial retrogradation of amylopectin [50,51]. This phenomenon can be observed in both starch and fat-starch systems. Changes occurring in fats and their gradual crystallisation undoubtedly caused an increase in the values of texture parameters, especially hardness (Table 3). These changes relate to the fats which are not bound in the form of amylose–lipid complexes [52].

Most studies on the rheomechanical and textural properties of low-fat emulsified meat products showed that partial replacement of fat with starch of various botanical origin, both native and chemically or physically modified, did not deteriorate but usually improved the functional properties of these products.

Garcia-Santos et al. investigated the influence of the replacement of fat with resistant starch in sausage on its physicochemical properties and sensory acceptance. The researchers observed that partial replacement of the adipose tissue with resistant starch up to 5% reduced the calorific value of the product, improved the stability of the emulsion, its colour parameters and texture profile. The modified sausages were accepted by consumers [53]. Pereira et al. investigated the effect of rice flour, glutinous rice flour, and tapioca starch on the stability of emulsion, moisture, protein secondary structure and the microstructure of cooked emulsified sausage. The study showed that among these exchangers replacing 15% of the fat content, tapioca starch had the greatest influence on the emulsion stability and cooking yield, and it contributed to greater firmness and more compact structure of the gel network [54]. Similarly, Rezler et al. [24] conducted a study to investigate how the substitution of fat with modified starch affected the quality of pork liver pâtés. The authors observed that the replacement of 5% of the fat content with modified potato starch preparations in the recipe composition of pâté stuffing significantly affected the texture of the final products. Regardless of the final temperature of thermal treatment, modified pâtés were characterised by lower hardness, spreadability, and adhesion than analogous unmodified products. The profile analysis showed that partial replacement of fat with modified starch improved the palatability of pâté sausages.

## 3. Materials and Methods

### 3.1. Materials and Sample Preparation

The research was conducted on fat-in-water emulsions prepared at a 1:3 fat:water ratio (g). They were supplemented with unmodified potato starch produced by Trzemeszno, Poland added at amounts of 1, 0.8, 0.6, 0.5, 0.4, and 0.2 of the fat mass. This corresponded to starch concentrations of 0.2, 0.17, 0.13, 0.11, 0.09, and 0.05 g/g, respectively. Starch concentration in the emulsion was defined as *m_s_*/(*m_s_* + *m_w_* + *m_f_*), where *m*_s_ was starch mass, *m_w_* was water mass and *m*_f_ was fat mass. Starch gels with the following concentrations corresponding to the concentration of starch in water were used as control systems in the emulsions: 0.25, 0.21, 0.17, 0.14, 0.12, and 0.06 g/g.

The chemical composition of starch was as follows: moisture, 13.2%; protein, 0.2%; fat, 0.1%; ash, 0.35%; amylose, 25%. Amylose content was determined by the iodine-binding procedure [55] through spectrophotometric detection (Shimadzu 2001).

Liquid porcine fat (Morliny, Poland) with a temperature of 40 °C was applied into the experimental system. Emulsion samples weighing 100 g were subjected to a 1 h thermal treatment at a constant temperature of 90 °C in an LWC2 M water bath (DANLAB Company, Grünwald, Germany). They were stirred continuously with an IKA Eurostar Power Control-Visc 6000 (IKA Werke GmbH, Staufen, Germany).

### 3.2. Rheomechanical Properties

The rheomechanical properties were measured with a DMWT dynamic mechanical thermal analyser (Cobrabid, Poznan, Poland). In vibration rheometers the viscoelastic properties and the examined systems are calculated by analysing and characterising the curve of the pendulum—free, damped vibrations with and without the sample (vibration frequency and the damping decrement [56]. A cone-plate measuring system was applied. The components of the complex modulus of elasticity *G*′ (storage modulus) and loss tangent (*tgδ*) were determined. *G’* is related with part of potential deformation energy which is maintained in the course of periodical deformations. The loss tangent (*tgδ*) is a measure of internal friction; it and determines the relative amount of energy dissipated in the material during one deformation cycle. The frequency of vibrations in the system was 1.2 Hz. The starch structuralisation kinetics was measure at a set temperature of 25 °C (with an accuracy up to 0.1 °C). Additionally, the starch gels and emulsions were measured after 24 h and 7 days of storage at 25 °C. The samples were prepared for tests as described in Section 3.1.

Rheomechanical tests were carried out at 25 °C. The linear viscoelastic region of each sample was considered. The rheomechanical tests of the samples were conducted in triplicate.

### 3.3. Composition of Fatty Acids

The following three fat extraction methods were used: the Folch method, the Rose-Gotlieb method, and the Soxhletmethod, with solvents of different polarity [57,58].

The composition of fatty acids in the porcine fat was determined by means of gas chromatography in accordance with the ISO/FDIS 17059 method (International Standard ISO/FDIS 17059 2007). A Hewlett-Packard 5890 SII apparatus equipped with a Supelcowax10 capillary column (30 m × 0.25 mm × 0.25 μm) and a flame ionisation detector (FID) was used for chromatographic separation. The analysis was conducted at a programmed furnace temperature ranging from 60 °C to 220 °C, with an increment of 12 °C per minute. The final temperature was maintained for 25 min [59]. The error of the method was 5%. The detection limit was 0.1%. Fatty acids were identified on the basis of the retention times of the standards.

### 3.4. Amylose Content and Complex Formation

The extent of the complex formation and the reduction in the iodine starch binding capacity of starch were investigated on the basis of the concept of complex index (*CI*) as described in other studies [36]. Deionized water (25 mL) was added into a 50 mL test-tube wherein gelatinised potato starch and a fatty mixture had already been prepared. The mixture was stirred for 2 min in a vortex. The dispersion was then centrifuged at about 1500× *g*, and 0.4 mL of the supernatant was mixed with 8.6 mL of deionized water and 1 mL of iodine solution (2.0% (*w*/*w* KI and 1.3% (*w*/*w*) I_2_ in deionized water) in a 15 mL tube. The absorbance (ABS) values of the sample and a reference (non-additive sample) were measured with a Shimadzu 2001 spectrophotomete at 690 nm. The CI was calculated from the following equation: *CI*, % = 100 × (ABS of reference-ABS of sample)/ABS of reference. All tests were performed in triplicate and the results were averaged.

### 3.5. Texture Analysis Method

Back extrusion was the method of measurement of texture with a TA-XT2i Texture Analyser (Surrey, UK) and A/BE type attachments. A sample weighing about 80 g was placed inside a cylinder with an internal diameter of 50 mm and compressed by a disc with a diameter of 40 mm to a depth of 30 mm. The use of a disc with a smaller diameter caused part of the emulsion sample to squeeze between the walls of the cylinder and the disc. During the compression and the return movement the disc moved at a speed of 1 mm/s. The instrumental measurement helped to calculate five mechanical determinants: surface destructive force (N), hardness (N), consistency (N∙s), cohesion (N), and the viscosity coefficient) (N∙s).

The following texture analyser settings were used:-head speed before compression: 1 mm/s,-head speed during compression: 1 mm/s,-head speed after compression: 1 mm/s,-disc penetration depth: 30 mm,-trigger force: 5 mm,-PPS (points per second): 250.

### 3.6. Sample Preparation for Texture Analysis

The texture was tested on starch–water and starch–water–fat systems containing native starch (Superior Standard) and pork fat. Double-distilled water was added to weighed starch according to the starch-to-water weight ratios given in Section 3.1. The resulting dispersions were kept in a water bath at a constant temperature of 95 °C for 60 min and continuously mechanically stirred at a speed of 500 rpm. The samples prepared in this way were placed in measuring vessels and then cooled down freely to room temperature (25 °C). Next, they were tested after one day and then after seven days of storage at room temperature (25 °C). Starch-water-fat systems were produced by adding pork fat liquefied at 40 °C to the starch dispersion according to the fat:starch:water weight ratios given in Section 3.1. Like the aqueous starch dispersions, the mixtures obtained in this way were kept in a water bath at a constant temperature of 95 °C for 60 min and continuously mechanically stirred at a speed of 500 rpm. The samples prepared in this way were placed in measuring vessels and then cooled down freely to room temperature (25 °C). Next, they were tested after one day and then after seven days of storage at room temperature (25 °C). The texture parameters of the samples were determined in triplicate

### 3.7. Statistical Analysis

The mean and standard deviations were calculated from the rheological measurements on three newly prepared samples and analysed using a spreadsheet statistical package (Microsoft Excel 2011). The values of changes of *k*(*c*)*, f, n* parameters were obtained by the method of computer fitting (least squares method, TableCurve 2D). One-way ANOVA was also used to determine the correlation between textural and rheological parameters (Statistica 10: Statsoft, Poland)

## 4. Conclusions

To determine the molecular rheomechanical and textural conditions resulting from the introduction of starch into the emulsion system of the animal (porcine) fat type in water, the time-related conditions of the course of the phenomena accompanying the process of formation of the spatial structure of starch and starch–fat systems were analysed.

The study showed that the rheomechanical and textural properties of the starch–fat gels resulted from conformational changes occurring over time within the structural elements forming their spatial network, which were conditioned by the concentration of starch and the presence of fat. Structural elements are segments formed by complex associates of amylose chains and by amylose–fat complexes. They form interpenetrating networks. Thanks to this, starch–fat gels are characterised by greater equilibrium concentration of the network segments than analogous starch gels. As a result, they have higher values of the storage modulus *G′* and higher values of the average functionalities of the terminal *f_k_* nodes. Despite this, the energy used for binding their structural elements to the network nodes is lower. In consequence, starch–fat systems have greater ability to dissipate mechanical energy (*tg**δ*) than starch gels.

The simultaneous instrumental investigations of the texture of analogous starch and starch–fat systems showed that the starch–fat systems had higher values of texture determinants than the starch systems. The high values of the correlation coefficients (*R*~0.9) between the texture determinants and rheological parameters, i.e., between hardness and storage modulus and between the index of viscosity and loss modulus, proved that the textural properties of the tested systems were strongly correlated with their rheomechanical properties. This suggests that both types of properties were conditioned by the same factors.

The knowledge of the mechanisms responsible for these properties will help to control the functionality of meat products, improve existing products, and help to develop new strategies of the production of sausages with a healthier profile (meat products with low fat content and a healthier lipid composition).

## Figures and Tables

**Figure 1 ijms-22-07276-f001:**
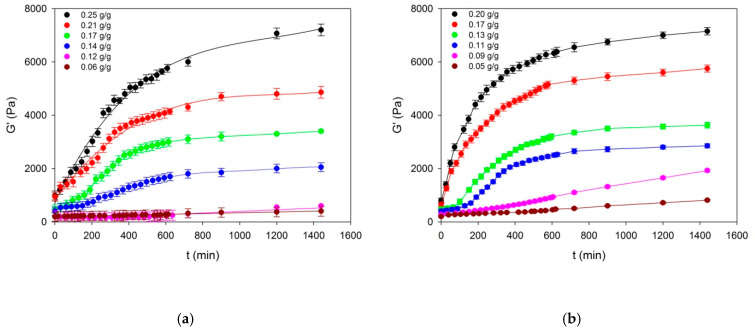
Kinetics of the storage modulus (*G’*) in the process of formation of starch gels (**a**) and starch–fat emulsions (**b**). The values are expressed as means ± SD (*n* = 3).

**Figure 2 ijms-22-07276-f002:**
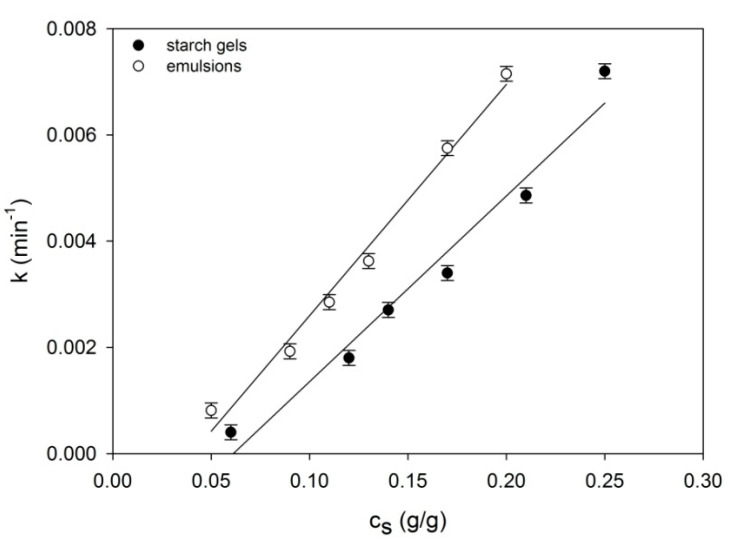
Concentrational relationships of the time constant *k* of the examined starch gels and starch-fat emulsions. The values are expressed as means ± SD (*n* = 3).

**Figure 3 ijms-22-07276-f003:**
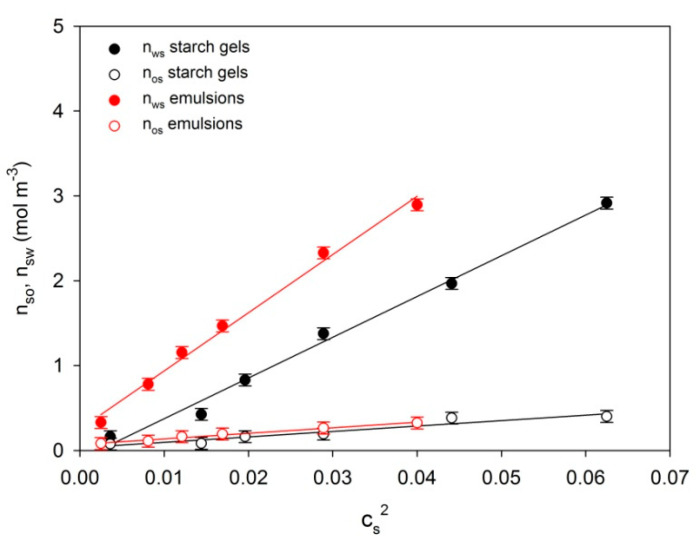
Concentrational relationships of initial *n_o_* and final *n_w_* concentrations of spatial network segments of the examined starch gels and starch–fat emulsions.

**Figure 4 ijms-22-07276-f004:**
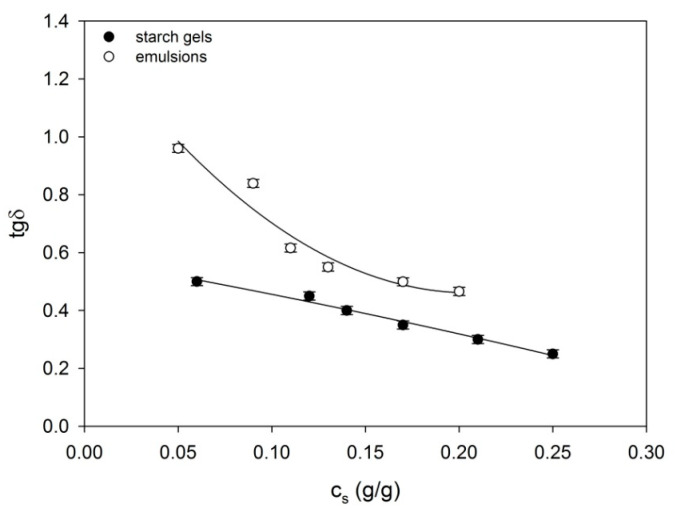
The *tg**δ* in the process of formation of the examined starch gels and starch–fat emulsions. The values are expressed as means ± SD (*n* = 3).

**Table 1 ijms-22-07276-t001:** The *CI* (%) values in the investigated starch–fat emulsions at different starch concentrations (g/g) in systems. The values are expressed as means ± SD (*n* = 3).

*c_s_*, g/g	CI, %
0.20	20.48 ± 0.89
0.17	27.64 ± 0.98
0.13	32.48 ± 0.74
0.11	40.20 ± 1.35
0.09	43.50 ± 1.12
0.05	45.24 ± 1.24

**Table 2 ijms-22-07276-t002:** Percentage content of fatty acids in the examined fats (porcine fat).

Fatty Acids	Content, %
C10:0	0.06
C12:0	0.07
C13:0	–
C14:0	1.37
C14:1 c-9	0.07
C15:0	0.12
C16:0	23.76
C16:1 c-9	2.34
C17:0	0.52
C18:0	17.86
C18:1 c-9	41.57
C18:2 c-9 c-12	7.38
C18:3 c-9 c-12 c-15	0.70
C20:0	0.23
C20:1 c-9	1.03

**Table 3 ijms-22-07276-t003:** The texture and rheomechanical parameters analysed of starch gels and starch–fat gels after one and seven days storage at temperature 25 °C. The below values are expressed as means ± SD (*n* = 3).

		*c_s_* _g/g_	Break Point N	Firmness N	Consistency N	Cohesiveness N	Index of Viscosity Ns	*G*′ Pas	*G*′′ Pas
after one day	starch gels	0.25	37.60 ± 0.90	69.49 ± 0.65	1659.09 ± 42.12	−24.04 ± 0.36	1254.93 ± 42.19	7200.8 ± 354.2	6001.2 ± 439.5
		0.21	32.32 ± 0.30	44.55 ± 0.43	1008.10 ± 32.17	−15.28 ± 0.22	725.80 ± 36.98	4860.0 ± 265.4	3075.4 ± 298.7
		0.17	9.01 ± 0.46	16.37 ± 0.70	334.57 ± 11.58	−5.10 ± 0.31	232.66 ± 8.49	3400.3 ± 189.1	2250.3 ± 192.1
		0.14	4.21 ± 0.21	12.10 ± 0.63	140.21 ± 4.34	−3.20 ± 0.15	113.24 ± 4.17	2050.1 ± 57.3	1325.6 ± 131.3
		0.12	1.18 ± 0.07	1.01 ± 0.05	30.07 ± 1.29	−0.31 ± 0.04	18.53 ± 0.54	600.8 ± 48.5	551.0 ± 32.8
		0.06	0.32 ± 0.01	0.52 ± 0.02	7.35 ± 0.43	−0.18 ± 0.02	4.11 ± 0.25	400.5 ± 11.7	350.6 ± 24.1
	starch-fat emulsions	0.20	20.51 ± 0.82	21.31 ± 0.85	583.67 ± 17.23	−9.54 ± 0.51	377.35 ± 10.93	7450.6 ± 345.1	6958.3 ± 248.7
		0.17	9.51 ± 0.35	11.92 ± 0.39	295.73 ± 6.96	−5.75 ± 0.17	171.25 ± 3.51	5750.5 ± 147.5	4763.1 ± 327.6
		0.13	1.41 ± 0.05	1.08 ± 0.02	110.47 ± 3.74	−1.35 ± 0.06	25.80 ± 1.39	3625.4 ± 125.7	3084.2 ± 211.4
		0.11	1.15 ± 0.04	0.76 ± 0.02	54.73 ± 2.01	−0.52 ± 0.05	13.42 ±0.73	2850.9 ± 123.4	2658.7 ± 190.1
		0.09	0.80 ± 0.03	0.75 ± 0.02	29.71 ± 1.61	−0.49 ± 0.04	8.34 ± 0.50	1925.3 ± 83.2	1899.1 ± 128.5
		0.05	1.10 ± 0.04	0.54 ± 0.01	11.19 ± 0.37	−0.10 ± 0.02	5.20 ± 0.04	812.5 ± 35.4	7396.8 ± 43.8
after seven day	starch gels	0.25	84.53 ± 4.07	159.28 ± 4.90	4126.50 ± 103.69	−22.62 ± 0.93	3830.44 ± 95.21	7800.7 ± 84.2	6240.5 ± 274.2
		0.21	49.72 ± 3.21	75.31 ± 3.84	2022.23 ± 77.15	−20.01 ± 0.84	1407.30 ± 51.37	5458.4 ± 59.3	4366.7 ± 271.3
		0.17	18.24 ± 1.11	16.48 ± 1.43	313.51 ± 16.39	−10.53 ± 0.41	213.45 ± 7.84	4190.6 ± 47.5	3352.4 ± 175.7
		0.14	4.11 ± 0.63	3.34 ± 0.39	87.26 ± 10.02	−4.51 ± 0.22	65.32 ± 2.79	3520.3 ± 31.1	2816.2 ± 145.3
		0.12	1.85 ± 0.27	1.11 ± 0.12	46.78 ± 7.78	−3.03 ± 0.19	26.38 ± 1.26	870.1 ± 11.5	696.0 ± 32.8
		0.06	1.12 ± 0.17	0.80 ± 0.09	23.02 ± 3.25	−0.6 ± 0.04	9.24 ± 0.68	600.9 ± 8.7	480.7 ± 21.4
	starch-fat emulsions	0.20	62.74 ± 4.57	57.91 ± 0.75	1750.58 ± 29.39	−14.78 ± 0.63	1610.40 ± 45.88	8469.6 ± 142.4	7775.6 ± 327.5
		0.17	26.62 ± 1.52	25.63 ± 0.46	773.85 ± 4.25	−12.62 ± 0.51	504.69 ± 10.24	6289.8 ± 131.7	5331.8 ± 269.7
		0.13	3.71 ± 0.21	2.94 ± 0.09	84.50 ± 2.65	−3.73 ± 0.18	44.38 ± 1.95	4293.7 ± 93.6	3434.9 ± 164.3
		0.11	2.93 ± 0.17	1.25 ± 0.05	68.73 ± 1.92	−0.80 ± 0.05	21.25 ± 0.91	3075.0 ± 74.4	2760.0 ± 136.1
		0.09	1.24 ± 0.09	0.99 ± 0.04	40.72 ± 1.46	−0.66 ± 0.04	18.80 ± 0.78	2147.2 ± 65.2	1917.7 ± 87.3
		0.05	1.30 ± 0.09	1.13 ± 0.05	25.09 ± 0.63	−0.11 ± 0.02	13.60 ± 0.66	1078.1 ± 30.5	862.4 ± 45.9

## Data Availability

The data presented in this study are available on request from the author.

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
