# Peer review of "DMA Study of the Molecular Structure of Porcine Fat in-Water Emulsions Stabilised by Potato Starch"

_ijms, 2021, doi:10.3390/ijms22147276_

Round 1
Reviewer 1 Report
Dear Author,
Revision is attached in the document.

Author Response
I find no sufficiently defined statement of hypothesis in the introduction, and therefore no in the conclusions as to whether your work confirmed or refuted your starting hypothesis. Also novelty and significance are unclear.
The Introduction has been expanded and the aim of the study has been better explained.
For example, please explain why Authors used porcine fat in examined emulsion ? This kind of fat is not nutritionally desirable.
Porcine fat was used in the model tests of emulsions because it, or rather the fatty tissue, is an important component of emulsified sausages, regardless of the type of meat it is made of (pork, beef or poultry). Since this type of fat is not healthy, it has been substituted with starches.
Other comments:
Introduction:
- Line 50 – needs citations or Please consider to put {13, 14} at the end this sentence.
Corrected
Results and Discussion:
- The acronym DMA should be explained first time (Line 62)
The acronym DMA is explained in the abstract.
- 2. Line 165 the expression “Porcine fat is made up,….” Is not appropriate please consider to use “contain” instead of “made up”
The sentence has been corrected.
- Regarding Table 2 fatty acid content should be expressed in two decimal places, not three. Also unsaturated fatty acids formula should include the position of double bond.foe example: C18:29c12c.
Corrected.
Materials and methods
The subchapter about analysis of fatty acid composition should be supplied with more information (standards, quantitative method).
The subchapter has been supplemented.
Reviewer 2 Report
This paper has not been submitted to a food science, hydrocolloid or meat science journal but rather to a general molecular structure journal. Therefore the Introduction requires far more background explanation. First and foremost, it is nearly impossible for all but a very specific group of readers to understand what this paper is about and why research is required. What exactly is the known structural effect of starch replacement on specific types of processed meats and what needs to be accomplished in order to improve textures of specific products? What exact types of meats are highly negatively affected by high levels of starch replacement? In fat-starch polymeric molecules, what types of bonding and starch/lipid choices are associated with better consumer acceptance and product quality?
Also this made even worse by the Journal article structure which places Material and Methods after Results and Discussion. Terms such as tgdelta and Cs are introduced in the body text with no definition and no explanation of why they matter. DMA is used in the Title, Abstract and Introduction with no definition. The same with G'--it is not defined until line 235! Further, the relevance of this parameter to the quality of the meat product and the stability, functionality etc. of the specific systems of low-fat processed meats is not explained. I have no idea after reading this carefully what starch concentration and starch/porcine lipid ratios the author believes would be preferable for food designs.
Fig. 1 has text superimposed on line numbers
Line 131 has "Figure" repeated
Line 148 No 1 sentence paragraphs please
174-174 Must be rewritten: English errors reduce understanding
Lines 209-211 Don't make sense and need rewriting
Both the Abstract and Conclusions are very poorly written and there is absolutely no clear explanation of how starch gels vs. starch/lipid gels actually behave. Most of the network bonding discussion regarding starch-starch and starch lipids seems to have been simply created out of speculations and other author's work and has no basis in any of the experiments presented in Material and Methods
The final conclusion Lines 327-330 makes no sense and is really not a Conclusion at all. Same with the last line of the Abstract. It is completely obvious with no research and to any layperson that starch emulsions would have different rheological properties than starch/lipid emulsions. Lipid is very different than starch. What is important is the quantified dose-response of these rheological changes with starch levels, and the application of this relationship to textural properties of processed meats. I believe the former topic was supposed to be covered in this manuscript but it is too disorganized and full of acronyms for me to be sure. As mentioned above, the latter topic is not mentioned at all.
Author Response
The Abstract has been changed. The Introduction has been expanded and the aim of the study has been better explained. The concepts of rheomechanical parameters and their meaning have been explained (the Results and Discussion have been supplemented). An analysis of the results of instrumental texture tests has been added. The Conclusions have been corrected. Lines 131, 148, 174-174, 209-211 have been corrected. Lines 327-330 have been deleted.
Information on the concentrations of starch emulsions and gels has been added to the Discussion.
Round 2
Reviewer 1 Report
The manuscript was sufficiently revised. In the present form I recommend it for publishing.
Reviewer 2 Report
I am pleased to see that my comments were seriously addressed, and now this paper has the structure, background, organization and correct scientific writing style to be a readable and useful contribution to the field. There are still quite a few spelling and grammatical errors and the manuscript needs a full edit by someone proficient in English. This is understandable given the extent of the text changes and I am confident that the authors and the journal editors working together can address this issue.